# Scavenging of Superoxide in Aprotic Solvents of Four Isoflavones That Mimic Superoxide Dismutase

**DOI:** 10.3390/ijms24043815

**Published:** 2023-02-14

**Authors:** Sandra Yu, Francesco Caruso, Stuart Belli, Miriam Rossi

**Affiliations:** Department of Chemistry, Vassar College, Poughkeepsie, NY 12604, USA

**Keywords:** superoxide, free radicals, superoxide dismutase, diabetes, cyclovoltammetry, formononetin, isoflavones

## Abstract

Isoflavones are plant-derived natural products commonly found in legumes that show a large spectrum of biomedical activities. A common antidiabetic remedy in traditional Chinese medicine, *Astragalus trimestris* L. contains the isoflavone formononetin (FMNT). Literature reports show that FMNT can increase insulin sensitivity and potentially target the peroxisome proliferator-activated receptor gamma, PPARγ, as a partial agonist. PPARγ is highly relevant for diabetes control and plays a major role in Type 2 diabetes mellitus development. In this study, we evaluate the biological role of FMNT, and three related isoflavones, genistein, daidzein and biochanin A, using several computational and experimental procedures. Our results reveal the FMNT X-ray crystal structure has strong intermolecular hydrogen bonding and stacking interactions which are useful for antioxidant action. Cyclovoltammetry rotating ring disk electrode (RRDE) measurements show that all four isoflavones behave in a similar manner when scavenging the superoxide radical. DFT calculations conclude that antioxidant activity is based on the familiar superoxide σ-scavenging mode involving hydrogen capture of ring-A H7(hydroxyl) as well as the π–π (polyphenol–superoxide) scavenging activity. These results suggest the possibility of their mimicking superoxide dismutase (SOD) action and help explain the ability of natural polyphenols to assist in lowering superoxide concentrations. The SOD metalloenzymes all dismutate O_2_^•−^ to H_2_O_2_ plus O_2_ through metal ion redox chemistry whereas these polyphenolic compounds do so through suitable hydrogen bonding and stacking intermolecular interactions. Additionally, docking calculations suggest FMNT can be a partial agonist of the PPARγ domain. Overall, our work confirms the efficacy in combining multidisciplinary approaches to provide insight into the mechanism of action of small molecule polyphenol antioxidants. Our findings promote the further exploration of other natural products, including those known to be effective in traditional Chinese medicine for potential drug design in diabetes research.

## 1. Introduction

Isoflavones, including formononetin (FMNT, 7-hydroxy-3-(4-methoxyphenyl)chromen-4-one), are bioactive phytochemicals that are abundant in legumes such as chickpeas, soy, beans and red clover [1]. These compounds are of nutritional and medicinal interest and are known as phytoestrogens because they bind with the β-estrogen receptor, a member of the nuclear hormone receptor (NHR) superfamily [2]. Additionally, FMNT is reported to have multiple biological activities, including neuroprotective, antitumor, antioxidant and anti-inflammatory effects in various in vitro and animal models [3,4,5]. Since literature reports show FMNT has moderate bioavailability, it is interesting to study for in vivo trials [6].

FMNT is also found in *Astragalus trimestris* L. (*Astragalus membranaceus Moench*) (*Fabaceae*), a common antidiabetic herbal remedy in traditional Chinese medicine [7]. Diabetes Mellitus is a serious global health issue, reaching a global prevalence of 10.5% across all adults aged 20–79 [8]. Insulin sensitivity is also promoted by the peroxisome proliferator-activated receptor gamma (PPARγ), a member of the NHR superfamily, and a ligand-activated transcription factor that also regulates adipogenesis. It is highly relevant for diabetes control and plays a major role in Type 2 diabetes mellitus development [9,10,11]. Current PPARγ-specific antidiabetics, the thiazolidinedione class of drugs such as rosiglitazone and pioglitazone, are full agonists and commonly increase insulin sensitivity. However, they have significant negative side effects, that limit their clinical indications and use [12]. In vitro and in vivo studies both show that FMNT can increase insulin sensitivity and potentially target PPARγ as a partial agonist [13,14,15,16]. FMNT is, therefore, a possible drug candidate that retains insulin sensitization ability without the unwanted side effects associated with full agonism, such as edema and weight gain.

To elucidate the biological role of FMNT, we performed several computational and experimental procedures: (1) single crystal X-ray crystallography to obtain FMNT’s molecular structure and information about its intermolecular interactions; (2) determination of FMNT antioxidant activity by measuring its scavenging ability of the superoxide radical; (3) use of computational DFT methods to understand the mechanism of FMNT scavenging of superoxide; (4) characterization of the docking interactions between PPARγ and FMNT using crystal-structure-obtained FMNT atomic coordinates. Upon observing our FMNT encouraging results, an equivalent analysis of superoxide activity was also performed for three related isoflavones: genistein, daidzein and biochanin A. FMNT and biochanin A are converted by 4′-O-demethylation through cytochrome P-450 enzymes to daidzein and genistein, respectively [17].

## 2. Results and Discussion

### 2.1. X-ray Diffraction

Single crystal X-ray diffraction data obtained on a suitable crystal of FMNT are reported in Table 1. The crystal structure shows hydrogen bonding and stacking interactions intermolecular interactions typical of polyphenolic compounds. Torsion angles show the chromone moiety rings, A and C, are in the same plane, while the exocyclic phenyl ring, designated B, is rotated −44.3(2)° from that plane as seen in Figure 1.

The FMNT crystal structure shows intermolecular interactions including a strong hydrogen bond (2.649 Å) between carbonyl O1 and hydroxyl group on C7 (Figure 2A). Other interactions listed in Table 2 include CH⋯O hydrogen bonds, as well as offset stacking interactions of 3.459 Å (Figure 2B).

In the Cambridge Structural Database (CSD) [18], there are crystallographic coordinates deposited for genistein (GENIST02), daidzein (XEKCUO) and biochanin A (IHAHIL). Investigation of these related structures shows that all of them have similar torsion angle twists of the exocyclic phenyl ring B with values of 47.71° and −45.92° for the two molecules in the asymmetric unit of biochanin A; 45.06° for daidzein; 54.97° for genistein. All three isoflavones also show offset stacking interactions at distances close to that seen in FMNT (3.453–3.485 Å).

### 2.2. Computational Antioxidant Activity

#### 2.2.1. FMNT Scavenging

Scavenging of superoxide was analyzed with DFT methods. In a recent review we described two ways of scavenging this radical by polyphenols: (1) superoxide interaction with an aromatic H(hydroxyl) (conventional polyphenol scavenging, that we call σ) and (2) interaction through the π–π approach [19], see Figure 1.

Figure 1 shows both scavenging modes for FMNT, which has only one H(hydroxyl) available for σ interaction, at position 7. The initial approach consists of posing one O(superoxide) atom at a van der Waals separation of 2.60 Å from H7, Figure 3A. After DFT geometry minimization, this H atom is captured by the radical to form the anion HO_2_^−^, while the remaining FMNT polyphenol comprises the unpaired electron located at ring A, Figure 3B. There are three options for π–π interaction, one for each of the A, B and C (pyrone) aromatic rings. The superoxide radical was posed over each ring at a van der Waals separation of 3.50 Å, between superoxide and ring centroids. Upon DFT optimization, the superoxide approaching ring A was directed towards H7, i.e., forming the same pattern shown in the σ attack. This is seen in the deposited Appendix A. When acting above rings B and C, the superoxide radical was rejected, as evidenced by final distances of 3.707 Å and 3.563 Å, respectively.

Scavenging details after the capture of H7 by superoxide FMNT are shown in Figure 3. Figure 3A shows the result after DFT minimization of X-ray FMNT coordinates, and a slight variation in the torsion angle between rings B and C is observed, −34.8° (DFT) and −44.7° (X-ray). Figure 3B shows the result of posing a van der Waals separated (2.60 Å) superoxide radical near H7; upon DFT minimization, H7 is captured with the formation of HO_2_^−^, separated 1.395 Å from O(7). When a proton is van der Waals posed near the more exposed O(superoxide) moiety, DFT minimization shows bond formation, O-H distance = 0.983 Å, while HO_2_^−^ is separated by 1.540 Å from the polyphenol. However, H7 results as returning to FMNT, O7-H7 = 1.044 Å, Figure 3C. This resonance stabilization through the phenolic ring A allows inclusion of the unpaired superoxide electron. From the configuration shown in Figure 3C, a second superoxide was placed over of ring A for π–π interaction at a van der Waals separation of 3.50 Å. At this point, the charge of the whole system is −1, due to two superoxide anion radicals plus one proton. Upon DFT minimization, H_2_O_2_ was formed and became well separated from ring A, 1.636 Å, while the stacked superoxide reagent became trapped within the ring, with a separation between centroids of 3.038 Å, which is shorter than the original van der Waals separation of 3.50 Å, Figure 3D. Hence, H_2_O_2_ was eliminated and upon DFT minimization, Figure 3E shows a partial double bond formation with a C7-O7 distance of 1.270 Å, which can be compared to the corresponding initial longer single bond moiety in FMNT, 1.379 Å, Figure 3A. Further comparison of Figure 3E with Figure 3A indicates that strong localization in ring A took place as C5-C6 (1.376 Å) and C8-C9 (1.382 Å) bonds are shorter than conjugated adjacent bonds, 1.427 Å, 1.421 Å, 1.454 Å and 1.467 Å. This alternating pattern of single–double bonds has been observed previously in related polyphenols after scavenging superoxide [20]. From the structure shown in Figure 3E, a proton was van der Waals posed near O7, and upon DFT optimization, FMNT reformed. However, the π–π interacting O_2_ molecule (O-O bond length of 1.258 Å) is still bound to ring A, as shown by a centroid separation of 3.015 Å, Figure 3F.

Next, we explored if this FMNT-η-O_2_ complex could behave as a catalyst for superoxide scavenging, i.e., beginning a new cycle. Another superoxide radical was σ van der Waals posed (2.60 Å) near H7, and DFT minimization (stopped after 81 cycles) showed initial formation of HO_2_^−^ ion, separated from FMNT 1.717 Å. More importantly, the π–π superoxide inserted in Figure 3D was rejected at a distance of 5.729 Å. Indeed, the latter is a molecule of O_2_, as shown by its O-O bond distance of 1.289 Å, which is much shorter than 1.373 Å for the superoxide. Thus, Figure 4 is formally equivalent to Figure 3A and confirms that FMNT can perform cyclic scavenging by consuming two superoxides, Figure 3B,D, plus two protons, Figure 3C,F, while giving H_2_O_2_, Figure 3D, and O_2_ Figure 4. This reaction (1) is the same as performed by superoxide dismutases (SOD), a family of metalloenzymes used to counteract excessive superoxide in cells [21,22]. Figure 2 shows the whole process of scavenging superoxide by FMNT.
2 O_2_^•−^ + 2H^+^ → O_2_ + H_2_O_2_(1)

#### 2.2.2. Genistein Scavenging

In Figure 3 we see that the scavenger genistein contains two additional hydroxyl groups in position 5 and 4′ compared to FMNT. The σ-scavenging of H7 (Figure 3) is identical to that shown by FMNT (H7 captured), Figure 5A. Additionally, the π–π interaction of superoxide with ring A induces a similar σ-scavenging of H7. In contrast with FMNT, the π–π interaction of superoxide onto pyrone ring C shows a minimum configuration where the superoxide is trapped, Appendix A, as shown by a centroid separation of 3.030 Å, i.e., forming a genistein-η-O_2_ complex. In addition, a H-bond is formed when superoxide σ approaches H in position 4′ of ring B, Appendix A, 1.625 Å. H in position 5 is not captured by superoxide, as the radical is rejected.

Figure 5A shows the σ attack on H7. Contrary to FMNT, when a proton interacts with the exposed O(superoxide) moiety of genistein-H7-O_2_, H_2_O_2_ forms and is well separated from genistein’s radical, 1.621 Å, which is not shown. Figure 5B shows the DFT minimization after superoxide is π–π added onto ring C. Figure 5C shows results from eliminating H_2_O_2_ and posing a proton near O7. Proceeding in a similar way as with FMNT, the additional superoxide posed near H7 induces the π–π molecule of O_2_ to leave, not shown, while the last superoxide forms a H-bond to H7. Meanwhile the second superoxide relocates above ring C, 3.154 Å, between centroids, Figure 5D. Thus, genistein behaves as FMNT and is able to perform another scavenging cycle, mimicking SOD action.

#### 2.2.3. Daidzein Scavenging

Figure 4 shows σ and π–π interactions between the superoxide and daidzein.

Daidzein behavior is similar to that of FMNT and genistein. Figure 6A shows the initial π–π attack of superoxide on ring A, which is equivalent to the previously described FMNT σ-scavenging on H7 that then forms HO_2_^−^. Next, a proton was posed near the most exposed O(HO_2_^−^) and DFT minimization resulted in H_2_O_2_ formation. Upon H_2_O_2_ elimination, a π–π superoxide was posed over ring A and DFT was applied resulting in the superoxide being released as a molecule of O_2_, Figure 6C. Finally, a proton was van der Waals posed near O7 and DFT calculations showed reformation of daidzein. In summary, daidzein is able to consume two superoxides, Figure 6A,C, plus two protons, one shown in Figure 6B, and the other in the process explained in Figure 6C, whereas H_2_O_2_ is eliminated after Figure 6B, and O_2_ is released after the final proton is incorporated in Figure 6C. Thus, daidzein, FMNT and genistein, are polyphenols able to act as SOD mimics.

#### 2.2.4. Biochanin A Scavenging

Figure 5 shows biochanin A results, where H7 is again σ scavenged. The π–π scavenging shows ring B rejection of superoxide and the formation of biochanin A-η-O_2_ complex with ring C centroid, 2.844 Å, as is the case with genistein, whereas the attack on ring A occurs as a σ scavenging on H7.

The biochanin A path of scavenging is similar to that of the other three isoflavones. First, H7 is well captured by superoxide, following an H atom transfer (HAT), both through σ or π–π approaches. Next, a proton was added, and upon DFT minimization, a well separated H_2_O_2_ moiety was seen, 1.621 Å from the remaining biochanin neutral radical. Hence, H_2_O_2_ was eliminated, the biochanin A radical was minimized and another proton was posed near O7. Upon DFT minimization, O7-H7 formed, and additional superoxide was a van der Waals π–π posed on ring A, 3.50 Å. The subsequent DFT minimized structure is the real catalyst for biochanin A mimicking SOD action, Figure 7.

### 2.3. Structural Details When the π–π Approach Is First Performed

As mentioned earlier, when the superoxide resides above ring A, the response of the four isoflavones is the same: the superoxide radical is redirected to H7 that is then captured, as shown in deposited Appendix A for FMNT. In contrast, there are differences in superoxide approaching the other two rings B and C. Thus, FMNT and biochanin A show rejection of the superoxide, whereas for genistein and daidzein, the superoxide is redirected towards the 4′-hydroxyl located in ring B, forming a H-bond. This is obviously not possible for FMNT and biochanin as they have a methoxy group in position 4′. The H-bond between superoxide and H4′(hydroxyl) is also formed after a π–π approach over ring B for genistein and daidzein. Interestingly, a superoxide π–π approach over the pyrone ring C results in a bond with genistein and biochanin being established, but not for FMNT and daidzein; the reason for this difference is not obvious. The difference in energy between reagents and products for the O_2_-η-biochanin-A complex, Figure 7, is −178.0 Kcal/mol, and for the equivalent O_2_-η-genistein, it is −147.7 Kcal/mol. Table 3 lists all possibilities of π–π interactions, with related structural parameters.

### 2.4. Hydrodynamic Cyclovoltammetry

The DFT scavenging activity of the four isoflavones was measured using a previously established cyclic voltammetry protocol [23]. This method uses a rotating ring disk electrode (RRDE) method that confirms and quantifies the antioxidant activity. Figure 8, Figure 9, Figure 10 and Figure 11 show the corresponding results for the four isoflavones in this study.

Table 4 shows slopes of Collection Efficiency (indicators of scavenging superoxide) [8] for several polyphenols using the RRDE method in the literature. The four isoflavones in the present study have slopes in between eriodictyol and butein (FMNT: −7.3 × 10^4^; genistein: −6.4 × 10^4^; daidzein: −7.0 × 10^4^; biochanin A: −7.1 × 10^4^). They are slightly weaker than DHDM, 2′,4′-dihydroxy-3,4-dimethoxy chalcone, which contains a 2′,4′-dihydroxy ring moiety responsible for scavenging. Butein, clovamide and quercetin are stronger scavengers than the isoflavones in this study and have catechol moieties, a structural feature associated with good scavenging activity. However, the best superoxide scavenger so far studied by the RRDE technique, galangin, has no ring B-hydroxyls.

### 2.5. FMNT Docking into the PPARγ Ligand Binding Domain (LBD)

To better understand the interactions of FMNT as a possible PPARγ partial agonist, atomic coordinates from the crystal structure of FMNT were docked in the PPARγ ligand binding domain (LBD), PDB Code: 5UGM [27], which contains edaglitazone in the active site. After applying CHARMm force field, the protocol “prepare protein” was applied to also provide H atoms to the protein. The edaglitazone position at the active site was selected to define the sphere of radius 10 Å and later eliminated. Docking of FMNT in this PPARγ LBD was affected for 10 poses and showed Cys285 π-interactions with FMNT rings. Pose 5 and pose 4 formed a cluster that was selected for further calculations, which included a standard dynamic cascade. The latter calculation confirmed Cys285 π-bonded to ring A (2.521 Å) and strengthened its interaction by the pyrone ring centroid, 2.758 Å (Figure 12). Initially, these were 2.569 Å and 3.110 Å, respectively, at docking. The environment of FMNT includes van der Waals interactions with Arg288 and Ile341 amino acids, Figure 13. Previous work has established that nearly every partial agonist interacts in a hydrophobic manner with Cys285 of Helix 3 and most interact with Arg288 using either electrostatic interactions or hydrophobic interactions. However, partial agonists that lack an acidic group can also stabilize the 𝛽-sheet by means of hydrophobic interactions, especially with the side chain of Ile341 [16]. These three amino acids show a similar role in our docking study, thus suggesting FMNT for potential use as partial agonist of PPARγ domain. Appendix A displays a partial view of the 5UGM protein, including FMNT pose 5 “Calculating Binding Energy” after standard dynamic cascade.

## 3. Materials and Methods

### 3.1. X-ray Structures

Formononetin was recrystallized from ethanol by slow evaporation. An APEX2 DUO platform X-ray diffractometer from Bruker Advanced X-ray Solutions was used to obtain X-ray data measurements at 125 K. Temperature was maintained using a cold liquid nitrogen stream from Oxford cryosystems; the X-ray source emitted MoKα radiation at 0.71069 Å. The crystal structure was solved and refined using full-matrix least-squares on F^2^ with the Bruker incorporated ShelX programs [28]. We input the X-ray data into the MERCURY program from Cambridge Structural Database (CSD) to produce images of the molecules and crystal packing [29]. Crystal data of FMNT have been deposited at the CSD and are available at https://www.ccdc.cam.ac.uk/structures/? (accessed on 28 October 2022) using Identifier CCDC number 2216211.

### 3.2. RRDE Measurement of Antioxidant Activity

Materials used to determine the antioxidant activity of the 4 isoflavones were tetrabutylammonium bromide (TBAB; Sigma Aldrich, St. Louis, MO, USA) and 99.9% anhydrous dimethyl sulfoxide (DMSO; Sigma Aldrich). The four isoflavones, FMNT, genistein, daidzein and biochanin A, were all obtained from Indofine Chemical Company (Hillsborough, NJ, USA). A 0.1 M TBAB/DMSO solution was used to produce electric current and enhance the occurrence of redox reactions. Antioxidant activity was measured via the hydrodynamic voltammetry technique with a rotating ring disk electrode (RRDE). The equipment used in this experiment was an MSR electrode rotator together with a WaveDriver 20 benchtop USB from Pine Instrumentation, Grove City, PA, USA. The main electrode tip was an E6RI ChangeDisk with a rigid gold ring and gold disk (Au/Au) insert. Before and after each experiment, 0.5 μm alumina suspension was used to clean the disk electrode tip (Allied High Tech Products, Inc., Rancho Dominguez, CA, USA) on a moistened polishing microcloth to eliminate potential film formation. A platinum (Pt) reference electrode and Pt counter electrode were also used in this experiment. All electrodes were obtained from Pine Research, Durham, NC, USA [30]. Cyclic voltammograms were run using a Solartron SI 1287 Potentiostat/galvanostat (Solartron Analytical, Oakridge, TN, USA) controlled through Coreware^©^ software. The antioxidant activity of the four isoflavones (all of them 99.9% pure) was determined based on their superoxide radical scavenging ability that was measured using the protocol developed in our lab [23]. Stock solutions of all compounds (99.9% purity) were prepared. Biochanin A, MW = 284.27, 0.071 g dissolved in 5 mL of DMSO, concentration = 0.05 M; genistein, MW = 270.25, 0.054g/4 mL DMSO, concentration = 0.05 M; formononetin, MW = 268.27, 0.107g/8 mL DMSO, concentration = 0.05 M; daidzein, MW = 254.24, 0.081 g/8 mL DMSO; concentration = 0.05M.

For the experiment, the electrolytic cell was bubbled for 5 min with a dry O_2_/N_2_ (35%/65%) gas mixture to establish its dissolved oxygen level. The Au disk electrode was then rotated at 1000 rpm while the disk was swept from 0.2 V to −1.2 Volts and the ring was held constant at 0.0 Volts, the disk voltage sweep rate was set to 25 mV/s. Several runs plus blank were performed in the RRDE experiment for the 4 compounds to determine their antioxidant activity with equal addition of 5 µL of each stock solution.

Results from each run were collected on Aftermath software and represented as voltammograms showing current vs. potential graphs that were later analyzed using Microsoft Excel. In an RRDE voltammetry experiment, the generation of the superoxide radicals occurs at the disk electrode while the oxidation of the residual superoxide radicals (that have not been scavenged by the scavenger) occurs at the ring electrode.
Reaction 2: Reduction of molecular oxygen at the disk electrode
Disk current O_2_ + e^−^ → O_2_^•−^(2)

Reverse Reaction 3: Oxidation of superoxide radicals at the ring electrode
Ring current O_2_^•−^ → O_2_ + e^−^(3)

Thus, the rate at which increasing concentrations of antioxidant scavenged the generated superoxide radicals during the electrolytic reaction was determined by obtaining the ring current/disk current (percent value) at each concentration. These values were denoted as the Efficiency of the scavenger at different concentrations. Using Microsoft Excel, Collection Efficiency values were plotted against the corresponding concentrations of each analyzed compound to produce a graph illustrating the effect of their increasing concentrations on the scavenging of superoxide radicals in the electrolytic solution. Ultimately, the slope of the curves served as a quantitative measure of the antioxidant activity of each compound.

### 3.3. Theoretical Calculations: Isoflavones Studied Using DFT and Molecular Mechanics

Calculations were performed using programs from Biovia (San Diego, CA, USA). Density functional theory (DFT) program DMol3 was applied to calculate energy, geometry and frequencies implemented in Materials Studio 7.0 [31]. We employed the double numerical polarized (DNP) basis set that included all the occupied atomic orbitals plus a second set of valence atomic orbitals, and polarized d-valence orbitals [32]; the correlation generalized gradient approximation (GGA) was applied including Becke exchange [33], plus BLYP correlation including Grimme’s correction when van der Waals interactions were involved [34]. All electrons were treated explicitly and the real space cutoff of 5 Å was imposed for numerical integration of the Hamiltonian matrix elements. The self-consistent field convergence criterion was set to the root mean square change in the electronic density to be less than 10^−6^ electron/Å^3^. The convergence criteria applied during geometry optimization were 2.72 × 10^−4^ eV for energy and 0.054 eV/Å for force. Calculations were generally performed with no solvent inclusion, those made in DMSO are specifically indicated. Docking studies were performed with the CDOCKER package in Discovery Studio 2020 version [35].

## 4. Conclusions

The X-ray structure presents FMNT’s ability to have strong intermolecular interactions, such as hydrogen bonding and stacking, which are useful for antioxidant actions. Antioxidant activity measured using the RRDE method revealed the four isoflavones to share a similar pattern in the collection efficiency slopes. All DFT calculations confirm these findings and allow us to conclude that antioxidant activity is based on the familiar superoxide σ-scavenging mode involving ring-A H7(hydroxyl) capture and supplemented by the π–π (polyphenol–superoxide) scavenging activity. These results suggest the possibility of their mimicking SOD action and help to explain the ability of natural polyphenols to assist in lowering superoxide concentrations. The SOD metalloenzymes all dismutate O_2_^•–^ to H_2_O_2_ plus O_2_ through metal ion redox chemistry [21], whereas these polyphenolic compounds do so through suitable intermolecular interactions (hydrogen bonding and stacking interactions). This theoretical assessment is confirmed using the cyclovoltammetry technique of rotating ring disk electrochemistry (RRDE), which results in very similar antioxidant activity for the four isoflavone scavengers, as shown by slopes in collection efficiency. Additionally, docking calculations suggest that FMNT can be a partial antagonist of the PPARγ domain. Our work shows the efficacy in combining multidisciplinary approaches, including experimental X-ray crystallography and RRDE, as well as computational docking and DFT methods, to provide an understanding of the mechanism of action of small molecule polyphenolic antioxidants. Finally, our findings promote the further exploration of other natural products, including those known to be effective in traditional Chinese medicine for potential drug design in diabetes research.

## Data Availability

Crystal data of FMNT have been deposited at the Cambridge Structural Database (CSD) and are available at https://www.ccdc.cam.ac.uk/structures/? (accessed on 28 October 2022) using Identifier CCDC number 2216211.

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
