# Peer review of "Scavenging of Superoxide in Aprotic Solvents of Four Isoflavones That Mimic Superoxide Dismutase"

_ijms, 2023, doi:10.3390/ijms24043815_

Round 1

Reviewer 1 Report

Review of the manuscript IJMS-2168849:

“Scavenging of superoxide in aprotic solvent of four isoflavones that mimic superoxide dismutase”

This paper combines experimental and theoretical work on four isoflavones identified as formononetin (FMNT), genistein, daidzein and biochanin A. However, I will only address the theoretical aspects reported that mainly deal with the mechanisms of the reactions between the isoflavones and the superoxide radical anion. Docking interactions involving FMNT are also studied.

The paper under review is well written but more work and better organization is needed before it is acceptable for publication. Some comments, suggestions and questions follow:

1.        Perhaps section 2.3 identified as “Computational Experiments” should be better identified as “Theoretical Calculations”?

2.     It seems the mechanistic theoretical study was performed in the gas phase and I do not think that is acceptable. Some approach for including solvent effects must be incorporated.

3.       Table 2: More details should be indicated in the heading and the table should not be broken between pages. This table relates to Figure 2, so perhaps its heading should indicate that the H-bond distances refer to the crystal structure. What are the numbers in parenthesis? What is listed under Donor-H and Acceptor-H? (All these should be better indicated)

4.     The computational work needs to be reported differently. Too many charts that repeat themselves a bit. Perhaps one general chart describing the possible pathways, but naming the reactions: HAT (hydrogen atom transfer), addition, single-electron transfer, etc). The results obtained must be better reported: no thermodynamic or kinetic data is tabulated for each reaction channel considered, and raw data is not provided. Calculated geometries of the various products and transition states must be reported with their standard enthalpy and Gibbs energy values.

5.      From the way the results are described, it does not seem that TS structures have been calculated. I do this type of work regularly and I am having trouble following the theoretical work reported here.

6.    Why study docking interactions with a crystal structure geometry when in solution geometries are most likely very different?

Author Response

Antioxidants : 4 isoflavones reviewer 1

Comments and Suggestions for Authors

Review of the manuscript IJMS-2168849:

“Scavenging of superoxide in aprotic solvent of four isoflavones that mimic superoxide dismutase”

This paper combines experimental and theoretical work on four isoflavones identified as formononetin (FMNT), genistein, daidzein and biochanin A. However, I will only address the theoretical aspects reported that mainly deal with the mechanisms of the reactions between the isoflavones and the superoxide radical anion. Docking interactions involving FMNT are also studied.

The paper under review is well written but more work and better organization is needed before it is acceptable for publication. Some comments, suggestions and questions follow:

  1. Perhaps section 2.3 identified as “Computational Experiments” should be better identified as “Theoretical Calculations”?

Done

  1.     It seems the mechanistic theoretical study was performed in the gas phase and I do not think that is acceptable. Some approach for including solvent effects must be incorporated.

 We added some examples of calculations including DMSO solvent effect, which is associated with the electrochemistry part of this study, and underlying their relevant structural differences with the gas phase results.  Figure 3D for FMNT, Figure 7A for daidzein, Table 3 for genistein and biochanin A.

  1.       Table 2: More details should be indicated in the heading and the table should not be broken between pages. This table relates to Figure 2, so perhaps its heading should indicate that the H-bond distances refer to the crystal structure. What are the numbers in parenthesis? What is listed under Donor-H and Acceptor-H? (All these should be better indicated)

 We modified the heading of Table 2 to say it refers to crystal structure shown in Fig 2 and the table is moved to be all together. Numbers in parentheses indicate the estimated standard deviation of these geometric data. (Since the atomic coordinates are experimentally found, they have estimated standard deviations.) Donor-H is the distance between the H bond Donor atom (D) and the hydrogen atom; and Acceptor-H is the distance between the H bond Acceptor atom  (A) and the hydrogen atom. This nomenclature arises from the geometric definition of hydrogen bonds widely and commonly used in crystal structure papers.

  1.     The computational work needs to be reported differently. Too many charts that repeat themselves a bit. Perhaps one general chart describing the possible pathways, but naming the reactions: HAT (hydrogen atom transfer), addition, single-electron transfer, etc.). The results obtained must be better reported: no thermodynamic or kinetic data is tabulated for each reaction channel considered, and raw data is not provided. Calculated geometries of the various products and transition states must be reported with their standard enthalpy and Gibbs energy values.

We have thought about how to best present the data and feel that one general chart that includes all 4 scavengers will be more difficult to understand. The isoflavones analyzed here follow a HAT pattern during the sigma attack. This is also supported by the electrochemistry study, which does not show peaks involving electron transfer to the scavengers. In addition, variation of energy, between reactants and products, involved in some of these processes have been included, one for each scavenger studied, Figure 3D for FMNT, Figure 7A for daidzein, Table 3 for genistein and biochanin A.  

  1.      From the way the results are described, it does not seem that TS structures have been calculated. I do this type of work regularly and I am having trouble following the theoretical work reported here.

Our group has reported transition states in other studies when needed. For example, a closely related example is when galangin H5 is captured by superoxide [Caruso, F. et al. PLoS ONE 2022, 17, e0267624]. Another case involves vitamin C, which releases its proton to an embelin derivative [Caruso, F. et al. Antioxidants 2020, 9, 382]. This is not the case here, as the reactions have no energy barriers. We thank the reviewer for this indication, and so we include now this sentence “calculations have no energy barriers in these geometry minimizations” in the manuscript, page 10, line 261; page 20, line 454.

  1.    Why study docking interactions with a crystal structure geometry when in solution geometries are most likely very different?

The X-ray input molecules are excellent model for analyzing reactivity in docking. It is widely accepted that the most likely observable crystal structures correspond to the lowest energy minima. In other words, if the crystal structure is available, it is useless to look for other sources. The docking protocol is a complex program, which takes the proposed initial structure, wherever it is obtained from (the crystal or another source), and randomly modifies it. In the case of FMNT, this is obtained, for instance, after variation in the torsion angle between B and C rings and so molecular rigidity in the crystal is not a problem. More generally, if a molecule contains many asymmetric carbons, the crystal structure will be the optimal starting point for docking, a related example is celastrol, https://doi.org/10.3390/ijms21239266]. Moreover, knowledge of crystal structure can suggest ways of interaction, mainly through H-bonds, which are energetically the most important components for interaction with amino acids, and these are very well observed in the crystal structure.

Reviewer 2 Report

The authors determined the X-ray structure of FMNT and found that its intermolecular interactions such as hydrogen bonding and stacking may be suitable for antioxidant action. They further used RRDE method for measurements and DFT for calculations. The work are well designed and experimentally performed. I suggest to accept this manuscript without further revision.

Author Response

We thank the reviewer for his/her positive opinio